# Atomic Layer Deposition Optimization via Targeted Adaptive Design

**Marieme Ngom**[1][*]  **Carlo Graziani**[1]  **Noah Paulson**[1]

[1]Argonne National Laboratory

[*]mngom@anl.gov

## Abstract

Atomic Layer Deposition (ALD) is a commonly employed process for producing conformal nanoscale coatings in the microelectronics and energy materials industries. ALD processes are composed of cycles of sequential self-limiting chemical reactions followed by purges with an inert gas to produce atomically thin coatings. At the end of each cycle, the Growth Per Cycle (GPC) which corresponds to net mass or thickness change from the previous ALD cycle is determined. Optimizing ALD processes for stable and uniform GPC for a new combination of reactants is challenging as the optimal combination of gas timings, temperature, and gas partial pressures spans a large multidimensional space and in-situ characterization is typically performed with a limited number of mass sensors. In this work, we use Targeted Adaptive Design (TAD), a Gaussian Process (GP)-based probabilistic machine learning framework that aims at efficiently and autonomously locating control parameters that would yield a desired target within specified tolerance, to optimize simulated ALD processes.

## 1 Introduction

Atomic Layer Deposition (ALD) [1] is a thin-film deposition technique that employs a cyclical process of alternating precursor exposures and purging steps, enabling self-limiting chemical reactions that deposit one atomic layer per cycle, providing precise control over film thickness and composition. In this work, we consider ALD processes that involve two chemicals, precursor 1 and precursor 2 that reacts cyclically with the surface of a substrate in a sequential and non-overlapping manner. Each cycle is composed of four sequences: a dose period $t_1$, during which the substrate is exposed to precursor 1; a purge period $t_2$, during which there is no precursor exposure; a dose period $t_3$, during which the substrate is exposed to precursor 2; and a final purge period $t_4$, during which there is no precursor exposure. At the end of each cycle, the GPC which corresponds to net mass or thickness change from the preceding ALD cycle is determined. In [5], the authors employed a simple physics model to simulate ALD and proposed three optimization methods- Bayesian Oprimization (BO) [6] with Expected Improvement [4], Random Optimization (*RO*) and Expert Systems Optimization (*ESO*) leveraging expert knowledge- to find timings $(t_1, t_2, t_3, t_4)$ that would yield Growth Per Cycle (GPC) saturation in minimum time. In this work, we use Targeted Adaptive Design (TAD) [2], a Gaussian Process (GP)-based method for efficiently and autonomously locating control parameters that yield a target feature within specified tolerances, to optimize ALD processes. We employ the physics model from [5] as a simulator and compare the performance of TAD to the methods proposed in that paper.

## 2 Overview of Targeted Adaptive Design

We start by giving a comprehensive overview of TAD which is a recently proposed algorithm for locating control parameters in a $D$-dimensional input space $X \subset \mathbb{R}^D$ that would yield a target feature $f_{target}$ in a $E$-dimensional output space $Y \subset \mathbb{R}^E$ within specified tolerance, while simultaneously

Workshop on Bayesian Decision-making and Uncertainty, 38th Conference on Neural Information Processing Systems (NeurIPS 2024).

learning the unknown mapping $f : X \mapsto Y$. The mapping $f$ is only attainable through noisy measurements $\boldsymbol{g} = f(\boldsymbol{x}) + \boldsymbol{\epsilon}$, where $\boldsymbol{x} \equiv \{x_k \in X : k = 1, \ldots, N\}$ and $\epsilon$ is a zero-mean Gaussian noise vector. TAD is an iterative method with similarities to BO, optimizing an acquisition function at each iteration to propose new candidate solutions.

## 2.1 Problem setup

TAD begins by defining three sets which it will update throughout its iterations: a set $\boldsymbol{x}_1$ of $N_1$ points in $X$ with associated noisy observations of $f$ $\boldsymbol{g}_1 \equiv f(\boldsymbol{x}_1) + \boldsymbol{\epsilon}_1$, a set $\boldsymbol{x}_2$ of $N_2$ points in $X$ associated with latent observations $\boldsymbol{g}_2 \equiv f(\boldsymbol{x}_2) + \boldsymbol{\epsilon}_2$, and a point $x \in X$ that corresponds to the initial target candidate solution. Additionally, TAD defines a Targeted Tolerance Region (TTR) $\eta_i \equiv [f_{target_i} - \tau_i, f_{target_i} + \tau_i]$, $i = 1, \ldots, E$, where $f_{target_i}$ is the $i$th component of the target vector $f_{target_i}$ and $\tau_i$ is a tolerance threshold for $f_{target_i}$. A vector-valued GP prior is assumed on $f$ i.e $f \equiv \mathcal{N}(\mu(.), C(., .))$ with $\mu$ and $C$ corresponding respectively the mean and covariance vectors of the GP. The GP is trained on the observations $(\boldsymbol{x}_1, \boldsymbol{g}_1)$ and is used to predict the distributions of $\boldsymbol{g}_2 | \boldsymbol{g}_1$ and of $f(x) | \boldsymbol{g}_1, \boldsymbol{g}_2$. In particular, we have

$$\boldsymbol{g}_2 | \boldsymbol{g}_1 \sim \mathcal{N}\left(\boldsymbol{p}^{(2|1)}, \boldsymbol{Q}^{(2|1)}\right) \tag{1}$$

$$\boldsymbol{p}^{(2|1)} \equiv \boldsymbol{\mu}_2 + \boldsymbol{K}_{21}\left(\boldsymbol{K}_{11} + \boldsymbol{\Sigma}_1\right)^{-1}\left(\boldsymbol{g}_1 - \boldsymbol{\mu}_1\right) \tag{2}$$

$$\boldsymbol{Q}^{(2|1)} \equiv \boldsymbol{K}_{22} + \boldsymbol{\Sigma}_2 - \boldsymbol{K}_{21}\left(\boldsymbol{K}_{11} + \boldsymbol{\Sigma}_1\right)^{-1}\boldsymbol{K}_{12}. \tag{3}$$

and

$$f(x) | (\boldsymbol{g}_1, \boldsymbol{g}_2) \sim \mathcal{N}\left\{\boldsymbol{p}^{(f(x)|1+2)}, \boldsymbol{Q}^{(f(x)|1+2)}\right\} \tag{4}$$

$$p^{(f(x)|1+2)} \equiv \mu(x) + \begin{bmatrix} \boldsymbol{K}_{x1} & \boldsymbol{K}_{x2} \end{bmatrix} \begin{bmatrix} \boldsymbol{K}_{11} + \boldsymbol{\Sigma}_1 & \boldsymbol{K}_{12} \\ \boldsymbol{K}_{21} & \boldsymbol{K}_{22} + \boldsymbol{\Sigma}_2 \end{bmatrix}^{-1} \begin{bmatrix} \boldsymbol{g}_1 - \boldsymbol{\mu}_1 \\ \boldsymbol{g}_2 - \boldsymbol{\mu}_2 \end{bmatrix} \tag{5}$$

$$Q^{(f(x)|1+2)} \equiv K_{xx} - \begin{bmatrix} \boldsymbol{K}_{x1} & \boldsymbol{K}_{x2} \end{bmatrix} \begin{bmatrix} \boldsymbol{K}_{11} + \boldsymbol{\Sigma}_1 & \boldsymbol{K}_{12} \\ \boldsymbol{K}_{21} & \boldsymbol{K}_{22} + \boldsymbol{\Sigma}_2 \end{bmatrix}^{-1} \begin{bmatrix} \boldsymbol{K}_{1x} \\ \boldsymbol{K}_{2x} \end{bmatrix}. \tag{6}$$

where $\boldsymbol{\mu}_1 \equiv \mu(\boldsymbol{x}_1)$, $\boldsymbol{\mu}_2 \equiv \mu(\boldsymbol{x}_2)$, $K_{xx} \equiv C(x, x)$, $\boldsymbol{K}_{x1} = \boldsymbol{K}_{1x}^T \equiv C(x, \boldsymbol{x}_1)$, $\boldsymbol{K}_{x2} = \boldsymbol{K}_{2x}^T \equiv C(x, \boldsymbol{x}_2)$, $K_{11} \equiv C(\boldsymbol{x}_1, \boldsymbol{x}_1)$, $K_{22} \equiv C(\boldsymbol{x}_2, \boldsymbol{x}_2)$, $K_{12} = K_{21}^T \equiv C(\boldsymbol{x}_1, \boldsymbol{x}_2)$, $\boldsymbol{\Sigma}_1$ is the noise covariance matrix associated to $\boldsymbol{\epsilon}_1$, and $\boldsymbol{\Sigma}_2$ is the noise covariance matrix associated to $\boldsymbol{\epsilon}_2$. The acquisition function is then constructed using these distributions.

## 2.2 Acquisition Function: Expected Log-Probability Density

TAD proposes a new acquisition function that computes the log-predictive probability density (LPPD) of the target design $f_{target}$ at the point $x$, conditioned on observations $(\boldsymbol{g}_1, \boldsymbol{g}_2)$. By Equation 4 and the standard formula for multivariate normal probability density, this is

$$\mathcal{L}_P(x, \boldsymbol{x}_1, \boldsymbol{g}_1, \boldsymbol{x}_2, \boldsymbol{g}_2) = -\frac{1}{2}\log\det Q^{(f(x)|1+2)}$$
$$-\frac{1}{2}\left(f_{target} - p^{(f(x)|1+2)}\right)^T \left(Q^{(f(x)|1+2)}\right)^{-1} \left(f_{target} - p^{(f(x)|1+2)}\right), \tag{7}$$

up to a constant. However, in this expression the mean $p^{(f(x)|1+2)}$ is dependent on the latent values $\boldsymbol{g}_2$ that are yet to be acquired. To obtain the TAD acquisition function, the expectation of the LPPD with respect to the predictive distribution $\boldsymbol{g}_2 | \boldsymbol{g}_1$ is calculated to obtain

$$\mathcal{L}_{TAD}(x, \boldsymbol{x}_1, \boldsymbol{g}_1, \boldsymbol{x}_2) \equiv E_{\boldsymbol{g}_2|\boldsymbol{g}_1}\left\{\mathcal{L}_\mathcal{P}(x, \boldsymbol{x}_1, \boldsymbol{g}_1, \boldsymbol{x}_2, \boldsymbol{g}_2)\right\}. \tag{8}$$

The TAD acquisition function $\mathcal{L}_{TAD}(x, \boldsymbol{x}_1, \boldsymbol{g}_1, \boldsymbol{x}_2)$ of Equation 8 has the following closed form:

$$\mathcal{L}_{TAD}(x, \boldsymbol{x}_1, \boldsymbol{g}_1, \boldsymbol{x}_2) = -\frac{1}{2}\log\det Q^{(f(x)|1+2)}-$$
$$\frac{1}{2}\left(f_T - p^{(f(x)|1)}\right)^T \left(Q^{(f(x)|1+2)}\right)^{-1}\left(f_T - p^{(f(x)|1)}\right)$$
$$-\frac{1}{2}\text{Trace}\left\{\left(\boldsymbol{K}_{x2} - \boldsymbol{K}_{x1}\left(\boldsymbol{K}_{11} + \boldsymbol{\Sigma}_1\right)^{-1}\boldsymbol{K}_{12}\right)\left(\boldsymbol{Q}^{(2|1)}\right)^{-1}\right.$$
$$\left.\times \left(\boldsymbol{K}_{2x} - \boldsymbol{K}_{21}\left(\boldsymbol{K}_{11} + \boldsymbol{\Sigma}_1\right)^{-1}\boldsymbol{K}_{1x}\right)\left(Q^{(f(x)|1+2)}\right)^{-1}\right\}. \tag{9}$$

TAD then defines two stopping rules: *convergence/success*, which occurs when a solution with uncertainties that fit within the TTR is found, and *convergence/failure*, which occurs when the search space is exhausted without finding such a solution (based on information theory). TAD iterates until one of these stopping conditions is met or the computational budget is reached. TAD thus offers an efficient search (with the $x_2$ points) at the cost of more function evaluations. A detailed discussion of the properties of this acquisition function, including its incorporation of the exploration/exploitation trade-off, as well as numerical simulations and comparisons with existing methods, can be found in [2].

## 3  Optimization of Atomic Layer Deposition (ALD):

The authors in [5] developed a physics-based model surrogate for ALD processes, which they sample with varying levels of noise. They then propose three different optimization methods, Bayesian Optimization with Expected Improvement (*BO/EI*) as an acquisition function, Random Optimization (*RO*) and Expert Systems Optimization (*ESO*) to find dose and purge timings that would yield GPC saturation in minimum time.

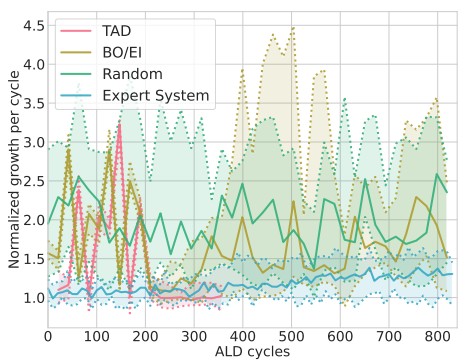
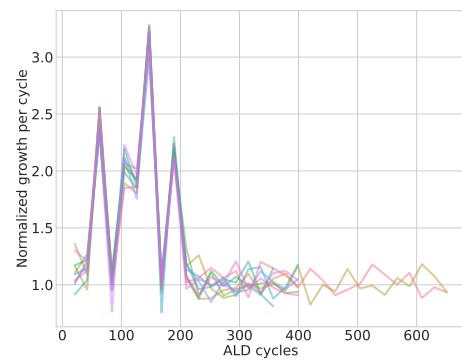

(a) TAD vs BO/EI, RO and ESO.  (b) TAD normalized GPC for all the different runs.

Figure 1: Comparison between our approach and current GP-based approaches

Prior to running the BO algorithm, an initial set of 10 observations is sampled using a Latin-hypercube design of experiments (LH-DOE) [3] to provide a representative starting point for the GP surrogate model. The objective function for *BO/EI* and *RO* is defined as a combination $C_{total}$ of four cost functions $C_1$, $C_2$, $C_3$ and $C_4$. $C_1$ focuses on minimizing the sum of the dose and purge timings relative to the sum of the maximum allowed time for each sequence. $C_2$ aims at reducing the divergence of the average GPC over multiple cycles from a reference GPC of a stable ALD process. $C_3$ aims at controlling the difference in GPC between an ALD process and one resulting from per-turbed timings while $C_4$ does the same n comparison to the initial GPC. The three different methods were run over 40 iterations for 10 different initializations of the physics model. We reproduce their experiments for the $Al_2O_3$ system at $200^oC$ and a noise value of 0.1. Figure 1a shows the mean and 95% confidence intervals of the normalized GPC $\hat{h}$ across these runs with respect to the number of ALD cycles (each iteration consists of 21 ALD cycles). *BO/EI* and *RO* produced timings that would yield desirable GPC but also explored values that led to excessive GPC during optimization. *ESO*, on the other hand, while reliable, suffered from overly conservative precursor dose times. Note that the variability during the first 210 ALD cycles for *BO/EI* is due to the LH-DOE sampling.

We now add TAD to the optimization strategies. We define the unknown function $f$ to be $f(t_1, t_2, t_3, t_4) = (C_{total}, \hat{h})$, with $C_{total}$ and $\hat{h}$ defined above, resulting in a 2-dimensional output space. We then set $f_{target} = (0.4, 1.)$ where $0.4$ is a conservative value derived from the *ESO* and $\hat{h}_{target} = 1$ ensures the normalized GPC remains stable around 1 during the optimization. We initialize the candidate solution to $(t_1, t_2, t_3, t_4) = (1, 1, 1, 1)$ and $x_2$ is initialized as in [2]. Finally, we set the tolerance for the TTR to be 10% across both output space dimensions. Since the GPC is dependent from previous ALD cycles, one needs to be conservative about the number $N_2$ of $x_2$ points to be acquired at each TAD iteration. Furthermore, the number of ALD cycles per TAD iteration is $N_2$ times more the number of ALD cycles per *BO/EI* iteration. To avoid having a prohibitive number of ALD cycles, we set $N_2 = 1$. We also run TAD for 10 different initializations

of the physics model and an observation noise level of $0.1$. The initial $(\boldsymbol{x}_1, \boldsymbol{g}_1)$ observations for the initial GP fitting are also constructed using the same LH-DOE procedure as before. If TAD reaches *convergence/success*, only the successful candidate solution is acquired after the last iteration. For each initialization, TAD successfully found a solution within the prescribed tolerance in $14$ to $21$ iterations (i.e. between $4$ to $11$ iterations after the $10$ initial LH-DOE steps) corresponding to $357$ to $651$ total ALD cycles, as illustrated in Figure 1. Furthermore, the normalized GPC remained stable across the $10$ different runs. Note that Figure 1a shows TAD results for the smallest number of iterations across the $10$ different initializations while Figure 1b shows TAD normalized GPC values for each of the runs over each corresponding number of iterations.

For illustration purposes, we show uptake curves which are commonly used graphical represen-

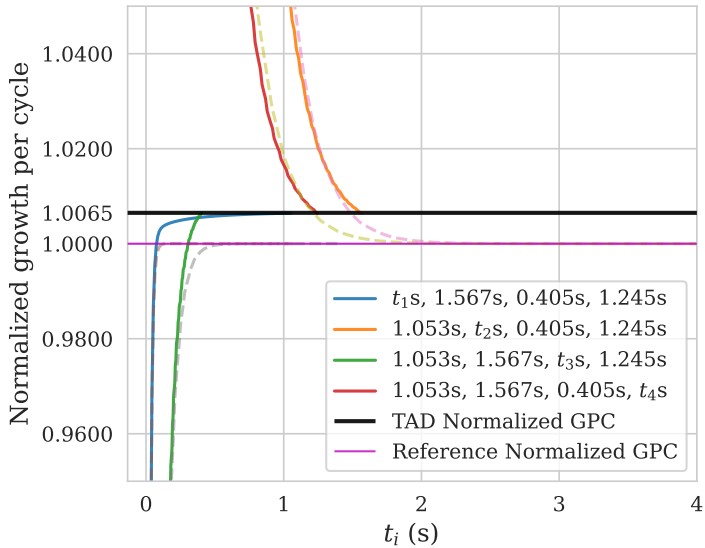

*Figure 2: The solid lines correspond to TAD uptake curves while the reference uptake curves for noise level $0.1$ are represented by the dashed lines.*

tations of the growth or deposition rate of material as a function of the precursor exposure time or dose. Uptake curves are obtained by varying one timing and fixing all the others at values resulting in saturation. Figure 2 shows uptake curves from one of the TAD runs (solid lines) superposed with reference uptakes curves for a noise level of $0.1$. TAD saturated slightly above the reference horizontal line $y = 1$ with saturation speed similar to the reference growth.

## 4   Discussion and future work:

We applied TAD, a recently developed batch iterative algorithm for locating control parameters that yield a target design, to the optimization of ALD. We compared TAD to existing optimization methods for ALD, including Bayesian optimization with expected improvement and have shown the effectiveness of our method in terms of the number of ALD cycles required for convergence and the stability of the normalized growth per cycle during optimization. In order to avoid a large number of ALD cycles per TAD iterations, we used a conservative number of points to be acquired at each TAD iteration, and aim to relax this constraint in the future by modifying the objective function and refining the function sampler require fewer ALD cycles per evaluation. Furthermore, we plan to analyze the effect of lower noise levels on TAD. In fact, in the zero-noise limit, the covariance matrices $Q^{(f(x)|1+2)}$ can become ill-posed especially in the presence of redundant $\boldsymbol{x}_2$ points. The authors in [2] proved that matrix has a finite limit when the noise goes to zero and it would be interesting to diagnose how TAD is affected in this case and compare with the other three methods presented here.

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
