# OpenReview forum: "Atomic Layer Deposition Optimization via Targeted Adaptive Design."
_NeurIPS.cc/2024/Workshop/BDU — NeurIPS BDU Workshop 2024 Poster_

### Official Review · Reviewer_ie1Y · 2024-09-20
**Accept – Optimizing Atomic Layer Deposition Using Targeted Adaptive Design**

**Rating:** 7
**Confidence:** 2

**Review:**

The paper presents a well-formulated optimization strategy for ALD processes using Targeted Adaptive Design (TAD). The proposed method improves upon existing optimization techniques, including Bayesian Optimization and Expert Systems Optimization, by offering a more efficient way to determine optimal control parameters for ALD cycles. The use of a Gaussian Process (GP)-based probabilistic framework enhances the quality of predictions, especially in the presence of noise, making it highly suitable for real-world applications.

In terms of quality, the methodology is sound and clearly articulated, with a solid theoretical foundation in probabilistic machine learning. The clarity of the paper is commendable, particularly in the explanation of TAD and its comparison to other optimization techniques. The visual aids, including graphs, effectively support the results.

The originality is notable, as TAD offers an innovative approach to optimizing ALD cycles, reducing both the number of cycles required for convergence and the variability in growth per cycle. The paper also makes significant contributions to the field of machine learning in industrial applications.

Cons: The method's limitations in lower-noise environments are acknowledged but could benefit from deeper exploration. Additionally, the scalability of TAD for more complex ALD systems needs further investigation.

---

### Official Review · Reviewer_Hp42 · 2024-10-07
**Optimize ALD via GP-based TAD**

**Rating:** 6
**Confidence:** 3

**Review:**

This paper applies a Gaussian Process (GP)-based probabilistic machine learning framework Targeted Adaptive Design (TAD) for optimizing the Atomic Layer Deposition (ALD). The optimization of ALD is challenging due to the multidimensional parameter space involved. The paper compares TAD with other methods such as Bayesian Optimization with Expected Improvement (BO/EI), Random Optimization (RO), and Expert Systems Optimization (ESO), showing that TAD-based ALD optimization achieves stable, optimized outcomes with fewer cycles.

Quality:
The quality of the paper is fair. It is technically sound and all claims are supported by the section 2 Overview of Targeted Adaptive Design (which I haven't reviewed in depth) are adopted from the original Targeted Adaptive Design paper. The final results on the ALD optimization is promising compared with other methods presented in this paper.

Clarity:
The paper is well written and easy to read. Despite some minor spelling error in the Conclusion, such as Q(f (x)|1+2 . Also, the link for citation [2] appears to be wrong.

Originality:
The paper is an application of using an existing algorithm TAD to optimize the ALD problem. The authors use this well-established GP-based techniques to solve a challenging problem that spans multidimensional space. In this regard, I do not think the method itself is particularly novel; However, the applied work resulting promising outcomes compared to other methods in the experiment section.

Significance:
Despite the potential practical significance of the work, I think the theoretical contributions are comparably small.

Pros:
- Apply the Targeted Adaptive Design (TAD) for optimizing ALD is an interesting approach.
- This work is focus on the practical usage by simulating and testing the current algorithm on optimizing ALD.

Weakness:
- It would be informative to see the optimization performance under different noisy level in the experiment section instead of focusing on the single set noisy level 0.1
- Add the statistical analysis. the authors are comparing the performance of TAD against other methods (e.g., BO/EI, RO, ESO) and have multiple experimental runs, statistical tests can be used to determine whether the observed differences in results are statistically significant or just due to random chance.

---

### Official Review · Reviewer_XiBN · 2024-10-07
**Review of Atomic Layer Deposition Optimization via Targeted Adaptive Design**

**Rating:** 4
**Confidence:** 4

**Review:**

This paper applies targeted adaptive design (TAD) to a material science problem. TAD is a recent method that aims to find a "configuration" $\mathbf{x}$ such that a function $f(\cdot)$ evaluates close to a "target region." To do so, TAD uses a certain expected log predictive density as an acquisition function. In the proposed application, TAD is applied to an atomic layer deposition problem and compared to Bayesian optimization, a random search, and an "expert systems optimization" appearing in the literature.

## Strengths
- I'm not an expert in TAD, but this seems like a reasonable application domain which TAD is designed for.
- While I'm skeptical of the loss function used for Bayesian optimization, the performance of TAD compared to the expert systems optimization method is promising.
## Weaknesses
- The paper can be somewhat difficult to read. The introduction to TAD is brief and lacks motivation, and some concepts are not defined (e.g., what "normalized GPC" is normalized w.r.t.).
- I'm a bit confused by the comparisons to Bayesian optimization. Results on the cost function $C_{\text{total}}$ are not shown, and instead, all comparisons are on normalized GPC, where Bayesian optimization is comparable to random sampling. Since the optimization problem is ostensibly smooth and low-dimensional, this seems suggestive of a poorly chosen loss function rather than a poor showing from Bayesian optimization.

  To this end, it seems more natural and fair to compare to Bayesian optimization with a loss function incorporating normalized GPC, since TAD is allowed to do this. For example, $C(x) = C_\text{total}(x) - \lambda (\hat{h}(x) - 1.0)^2$ for some constant $\lambda$. This also sidesteps the issue of defining a reasonable value of $C_\text{total}$ to imitate, like is necessary in TAD.

There are also some small typographical or editorial issues. Here are a few suggestions I would encourage the authors to incorporate:
- All acronyms should be defined in the text. For example, GPC is only defined in the abstract, and LPPDD is not defined at all.
- I think $\mathcal{L}(\cdot)$ is chosen because the acquisition function is a log-likelihood, but it is a somewhat unfortunate notation to maximize a quantity called $\mathcal{L}(\cdot)$ in an optimization paper. I'd encourage the authors to find an alternative notation, or at least highlight that $\mathcal{L}(\cdot)$ is maximized.
- Vector/matrix notation should be consistent, e.g. $x$ is inconsistently bolded throughout the text, and $\mathbf{p}$ and $\mathbf{Q}$ are inconsistently bolded between Eq. (1) and Eq. (4).
- "tp" should probably be "to" in the first paragraph of Section 3, and $Q^{(f(x) | 1 + 2)}$ is missing the closing parenthesis in Section 4.
- The title on OpenReview has a typo ("layed" instead of "layer").
- Figure 1 is not readable without colors (e.g., when printed in grayscale). It would be great if the figure is modified with symbols to be more readable in grayscale.

## References
[1] Graziani, C., & Ngom, M. (2024). Targeted Adaptive Design. _arXiv preprint arXiv:2205.14208_.

---

### Decision · Program_Chairs · 2024-10-09

Accept (Poster)